# Invasive Aspergillosis by *Aspergillus flavus*: Epidemiology, Diagnosis, Antifungal Resistance, and Management

**DOI:** 10.3390/jof5030055

**Published:** 2019-07-01

**Authors:** Shivaprakash M. Rudramurthy, Raees A. Paul, Arunaloke Chakrabarti, Johan W. Mouton, Jacques F. Meis

**Affiliations:** 1Department of Medical Microbiology, Postgraduate Institute of Medical Education and Research, Research, Chandigarh 160012, India; 2Department of Medical Microbiology and Infectious Diseases, Erasmus MC, 3015GD Rotterdam, The Netherlands; 3Department of Medical Microbiology and Infectious Diseases, Canisius Wilhelmina Hospital (CWZ) and Center of Expertise, 6532SZ Nijmegen, The Netherlands; 4Center of Expertise in Mycology Radboudumc/CWZ, 6532SZ Nijmegen, The Netherlands

**Keywords:** invasive aspergillosis, *Aspergillus flavus*, epidemiology, molecular typing, azole resistance, amphotericin B resistance, treatment, epidemiological cut-off value

## Abstract

*Aspergillus flavus* is the second most common etiological agent of invasive aspergillosis (IA) after *A. fumigatus*. However, most literature describes IA in relation to *A. fumigatus* or together with other *Aspergillus* species. Certain differences exist in IA caused by *A. flavus* and *A. fumigatus* and studies on *A. flavus* infections are increasing. Hence, we performed a comprehensive updated review on IA due to *A. flavus*. *A. flavus* is the cause of a broad spectrum of human diseases predominantly in Asia, the Middle East, and Africa possibly due to its ability to survive better in hot and arid climatic conditions compared to other *Aspergillus* spp. Worldwide, ~10% of cases of bronchopulmonary aspergillosis are caused by *A. flavus.* Outbreaks have usually been associated with construction activities as invasive pulmonary aspergillosis in immunocompromised patients and cutaneous, subcutaneous, and mucosal forms in immunocompetent individuals. Multilocus microsatellite typing is well standardized to differentiate *A. flavus* isolates into different clades. *A. flavus* is intrinsically resistant to polyenes. In contrast to *A. fumigatus*, triazole resistance infrequently occurs in *A. flavus* and is associated with mutations in the *cyp51C* gene. Overexpression of efflux pumps in non-wildtype strains lacking mutations in the *cyp51* gene can also lead to high voriconazole minimum inhibitory concentrations. Voriconazole remains the drug of choice for treatment, and amphotericin B should be avoided. Primary therapy with echinocandins is not the first choice but the combination with voriconazole or as monotherapy may be used when the azoles and amphotericin B are contraindicated.

## 1. Introduction

Invasive aspergillosis (IA) is generally encountered in immunocompromised patients with steroid treatment, chemotherapy resulting in severe neutropenia, hematopoietic stem cell, and solid organ transplantation. IA has a high mortality rate and *Aspergillus fumigatus*, *A. flavus*, *A. niger*, *A. terreus*, and *A. versicolor* are the most common species involved. The genus *Aspergillus* encompasses more than 250 species and is one of the largest genera of filamentous fungi causing human diseases [1,2]. Worldwide, *A. fumigatus* is the most common agent of invasive aspergillosis and has been widely studied and reviewed. Infection due to *A. flavus* is predominant in Asia, the Middle East and Africa possibly due to its better ability to survive in hot and arid climatic conditions compared to other *Aspergillus* spp. *A. flavus* causes clinical syndromes similar to *A. fumigatus* in humans [3]. Experimental in-vivo studies have shown that *A. flavus* is more virulent than *A. fumigatus* and other *Aspergilli* in terms of the time and initial inoculum required in causing mortality in both normal and immunocompromised experimental mice [4]. Other than IA, diseases due to *A. flavus* manifest in various forms including allergic syndromes and saprophytic colonization of cavities and sinuses [5,6]. Of the world’s ten most feared fungi, *A. flavus* has been placed in the fifth rank because, in addition to human diseases, it also causes pre- and post-harvest diseases in several crops and aflatoxin-related toxicities in humans and animals [7]. *A. flavus* has been demonstrated to differ from *A. fumigatus* in terms of geographical distribution, pathogenic potential and antifungal resistance profile [6]. The present review aimed to update the available literature on the epidemiology, antifungal resistance, diagnosis, and management of IA due to *A. flavus*. 

## 2. Literature Review

The literature review for this study consisted of a search in Medline through PubMed, and the Cochrane Library databases using various combinations of key words such as invasive aspergillosis and *A. flavus*, epidemiology, prevalence, diagnosis, galactomannan, beta-d-glucan, antifungals, resistance, resistance mechanism, epidemiological cut-off values, management, and treatment. Only the English language literature and all published studies up until April 2019 relevant to the aim were selected and reviewed.

## 3. Clinical Spectrum and Distribution

*Aspergillus* species are extraordinary in the context of the diversity of its clinical manifestations. Perhaps no other human infectious agent has such a wide clinical spectrum. All forms of infections are transmitted from the abiotic environment (sapronoses) and are often not communicable from person to person. However, recently, it has been demonstrated that *A. fumigatus* can produce aerosols and has the potential to transmit to other persons [8]. Climatic and geographic conditions may be important determinants of the local prevalence and distribution of *Aspergillus* species. *A. flavus* is more prevalent in the environment of some tropical countries like India, Mexico, Pakistan, Sudan, and Saudi Arabia and consequently remains the most frequent species causing aspergillosis in those countries [9,10,11,12,13,14]. To understand the reason for this differential geographical distribution of *Aspergillus* species, further studies on underlying biological attributes of different *Aspergillus* species are warranted. 

The spectrum of aspergillosis is broadly classified into four categories: invasive life-threatening infections in immunocompromised persons; sub-acute or chronic infections in patients with structural lung abnormalities or pre-existing pulmonary or sinus disease or some subtle defect in innate immunity; allergic or eosinophilic disease manifested in many forms like allergic bronchopulmonary aspergillosis (ABPA), eosinophilic rhinosinusitis, and extrinsic allergic alveolitis; and locally invasive infections as a result of trauma or surgery such as keratitis or post-operative infections.

### 3.1. Invasive Aspergillosis

Invasive aspergillosis implies intrusion of *Aspergillus* hyphae into tissues which is discernible on histological examination. The most common organ affected is the lungs, followed by paranasal sinuses and the central nervous system. The disease primarily affects patients with neutropenia, which serves as the classical risk factor for invasive aspergillosis. However, it can also affect non-neutropenic immunocompromised patients and even critically ill immunocompetent hosts. Recently, Chakrabarti et al. conducted a multicentric study on invasive mold infection in Indian ICUs and reported *Aspergillus* (47%—*A. flavus* vs. 39.4%—*A. fumigatus*) as the most common fungus isolated from patients with non-classical risk factors (63.5%) surpassing the classical risk factors (36.4%) [13]. More recently, it has been observed that IA can develop concurrently with severe influenza in apparently immunocompetent individuals [13,15,16,17]. In a series of 18 cases of influenza associated aspergillosis (IAA) from China, *A. flavus* was implicated in three patients (17%), two survived, and one expired [17]. In the multicentric study on ICU acquired mold infection from India, 142 cases of invasive aspergillosis were observed of which 12 (8.5%) cases were IAA and half of those were due to *A. flavus* [13]. *A. flavus* is associated with sino-orbital aspergillosis and ocular infection, especially in developing countries [5]. Leug et al. reported a cluster of eight invasive fungal sinusitis cases which were directly related to increased airborne conidial counts after soil excavation during hospital renovation. In six of those cases, the etiological agent was *A. flavus* [18].

It has been observed that increase in mean concentrations of *A. fumigatus* and *A. flavus* conidia from <0.2 to >1 conidium per cubic meter of air lead to an increased number of cases of invasive aspergillosis from 0.3%–1.2% in immunosuppressed patients [19]. Saghrouni et al. reported a high isolation rate of *A. flavus* (73.7%) from invasive pulmonary disease in neutropenic patients in Tunisia [20]. 

#### 3.1.1. Pulmonary Aspergillosis

The term pulmonary aspergillosis encompasses both invasive as well as non-invasive categories of infections and can be classified into invasive pulmonary aspergillosis (acute and chronic), semi-invasive aspergillosis, pulmonary aspergilloma, or allergic bronchopulmonary aspergillosis. Approximately 10% of cases of bronchopulmonary infections are caused by *A. flavus*, whereas *A. fumigatus* accounts for the majority of cases of pulmonary aspergillosis [3]. Very few cases of chronic cavitary pulmonary aspergillosis and aspergilloma are associated with *A. flavus* and are mainly reported from areas with hot and dry climates [21]. The low incidence of *A. flavus* in causing pulmonary disease may be related to restricted entry and germination of conidia in the alveoli due to their relatively larger size and invasion in the human milieu as compared to *A. fumigatus* [6,22]. Zarrinfar et al. showed a higher isolation rate of *A. flavus* (39%–54%) from bronchoalveolar lavage (BAL) samples of patients with pulmonary or respiratory disorders and solid organ transplant patients, thereby depicting a high prevalence of *A. flavus* colonization or infection in these patients [23]. Most lung aspergillomas are reported to be located in the upper lobes in prior cavitary lesions. Aspergillomas in lungs due to *A. flavus* may be multiple or bilateral and are rarely associated with pneumothorax [12,24]. 

#### 3.1.2. CNS Aspergillosis

*A. flavus* can involve the brain as an extension of infection from a primary lesion in the nasal and paranasal sinuses, mastoid bone, or middle ear in immunocompetent hosts or through hematogenous spread as a part of dissemination in an immunocompromised host [5,25]. The rhino-cerebral form of CNS aspergillosis is the commonest form as this entity can be diagnosed early because of ease of sampling from the paranasal sinuses [25]. These patients have low mortality due to early debridement and prompt antifungal therapy. This form is more commonly found in developing countries, as paranasal fungal infections are more prevalent in Asia, the Middle East, and Africa [5]. CNS aspergillosis which manifests as an intracerebral abscess after hematogenous dissemination in immunocompromised patients often remains under-diagnosed because of lack of any typical diagnostic characteristic and difficulty in sampling. A majority of patients succumb to this type of infection and are diagnosed at autopsy [26,27]. In the western USA, 24% of CNS aspergillosis in organ transplant patients was caused by *A. flavus* [28]. Most cases of neuro-aspergillosis due to *A. flavus* have been reported from India, Pakistan, the Middle East, and Africa [29].

#### 3.1.3. Endophthalmitis Due to *A. flavus*

Endophthalmitis due to *Aspergillus* spp. is usually associated with post-operative or post-traumatic risk factors. In a single center from northern India, the most common agent of fungal endophthalmitis was *Aspergillus* (54.4%), and *A. flavus* alone accounted for 24.6% of all the fungal endophthalmitis cases [30]. In another study of 27 cases of post-cataract surgery endophthalmitis, *A. flavus* was isolated in 59% of the cases [31]. During a single center study from southern India, *A. flavus* was most commonly implicated in this disease [32].

#### 3.1.4. Invasive Fungal Rhinosinusitis

Invasive fungal rhinosinusitis is classified into acute invasive (AIFRS), chronic invasive (CIFRS), and chronic granulomatous invasive fungal rhinosinusitis (CGIFRS) [33]. The acute (fulminant) invasive (AIRS) type most commonly afflicts immunosuppressed patients and presents with a short history of <4 weeks. This disease is characterized by vascular invasion and necrotizing tissue reaction with abundant hyphae. Though the magnitude of AIFRS is almost similar in the developed and developing world, risk factors and etiological agents involved varies. *A. fumigatus* is the most common etiological agent in developed countries, but in developing nations, cases due to *A. flavus* have been increasingly noted [5]. Michael et al. from southern India showed *A. flavus* (10/51 = 19.6%) as the second most common etiologic agent of AIRS after *Rhizopus oryzae* (29/51 = 56.8%) [34]. CIFRS occurs mostly in mildly immunosuppressed patients, including diabetes mellitus, HIV infection, or those on corticosteroid therapy. This disease is more indolent in its behavior and lasts for more than 12 weeks with a relatively slow progression. This entity is most commonly seen in Western countries and Japan [35]. Whereas CGIFRS is usually encountered in immunocompetent individuals, tissue invasion remains largely local, affecting the nose, cheek, paranasal sinuses, and orbit, often accompanied with proptosis. The invasion may progress to involve bone erosion and tissue destruction extending to the brain, cavernous sinuses, and major vessels [36]. *A. flavus* is almost exclusively the causative agent reported from India, Pakistan, Saudi Arabia, and Sudan [37,38].

#### 3.1.5. Cardiac Aspergillosis

Cardiac aspergillosis, although rare, is mainly reported following cardiac surgery. *A. flavus* has been implicated in 11.2% of cases, and most of them were associated with contaminated grafts, contaminated sutures, or intra-operative dispersion of conidia [39]. The infection may present as endocarditis, aortitis, the involvement of pacemaker, and pericardium [39,40]. Both native and prosthetic valves can be affected by *A. flavus* [39]. Brili et al. reported a case of *A. flavus* infection of an ascending aortic aneurysm after cardiac surgery in a diabetic patient [41]. In a case of post-operative endocarditis from Spain, *A. flavus* strains from the heat exchanger and the one from the graft showed 100% concordance indicating a hospital-acquired infection [42]. In a review of *Aspergillus* endocarditis following transplantation from 1975–2017, of 28 cases identified, two cases were due to *A. flavus* [43].

#### 3.1.6. Cutaneous and Subcutaneous Aspergillosis

Wound infections due to *A. flavus* usually affect central venous catheter insertion site or causes secondary infection due to hematogenous spread. The lesion may present as macules, papules, bullae, nodules, ulcers, and abscesses [3,39]. In a multicentric study from France, all cases of primary cutaneous aspergillosis were due to *A. flavus* [44]. Post-operative wound infections may be associated with high concentrations of aerial spores in the operation theatre [39]. *A. flavus* is also reported to cause fatal aspergillosis in neonates, sternal wound infections after cardiac surgery, and stem cell transplantation [45,46]. Subcutaneous aspergillosis is a very rare phenomenon, and the lesions can arise either by primary traumatic inoculation of *Aspergillus* or as a manifestation of disseminated aspergillosis. The most common clinical form of primary subcutaneous aspergillosis is eumycetoma characterized by swelling, draining sinuses, and granules. Until now, five cases of eumycetoma due to *A. flavus* have been reported, two from Sudan and one each from the USA, Iran, and south India [27,47,48,49].

#### 3.1.7. Bone and Joint Infections

Among the complications of invasive aspergillosis, the musculoskeletal manifestation of invasive aspergillosis is a rare phenomenon. *A. flavus* is documented to be an important cause of osteomyelitis after trauma [29]. It is also responsible for deep sternal wound infections, costochondritis, and osteomyelitis of the ribs and chest wall, 3–8 months following cardiac surgery. In a review of 310 patients with *Aspergillus* osteomyelitis, *A. flavus* was implicated in 12% of cases [50]. The chronic granulomatous disease has also been found to be an important risk factor for *A. flavus* infection of vertebrae. In another review by Koehler et al., *A. flavus* was implicated in 18% of IA cases including mastoiditis, discitis, vertebral osteomyelitis, septic arthritis of the shoulder, skull base osteomyelitis, and epidural abscess [51]. *Aspergillus* arthritis usually develops as a secondary infection in disseminated aspergillosis after hematogenous spread. Patients present with complaints of edema and pain of knee joints, intervertebral discs, and hip joints. In a series of 31 cases of *Aspergillus* arthritis, *A. fumigatus* was isolated from 77% of the cases followed by *A. flavus* in 13% of which 52% of those cases were disseminated aspergillosis while 39% developed infection after direct inoculation [52].

## 4. Diagnosis

The diagnosis of invasive aspergillosis by any species of *Aspergillus* is generally similar [3]. However, there is a need to consider a few factors while diagnosing aspergillosis caused by *A. flavus*. The standard approach for the diagnosis of proven invasive aspergillosis is demonstration of septate acute angle branching hyaline hyphae followed by isolation and identification from tissues. In pulmonary aspergillosis, it has been shown that the burden of *A. flavus* in lung tissue is generally higher than in bronchoalveolar lavage (BAL) samples suggesting that an invasive tissue biopsy could be a better sample than BAL for diagnosis of invasive disease due to *A. flavus* [3]. IA can be diagnosed using different biomarkers such as galactomannan (GM) and (1→3)-β-d-glucan detection in serum and BAL and by applying molecular techniques [3]. An in-vitro study by Swanink et al. showed a 7% higher production of GM by *A. flavus* as compared to *A. fumigatus* [53]. An in-vitro study by Xavier et al. also showed a lower release of GM by *A. flavus* compared to *A. fumigatus* [54]. In a study of hematologic malignancy patients by Hachem et al., the sensitivity of serum GM was higher with aspergillosis due to non-*fumigatus Aspergillus* species (49%) than those infected with *A. fumigatus* (13%), though non-*fumigatus Aspergillus* species were not specified in the study [55]. The mean BAL-GM index in pulmonary aspergillosis patients with *A. flavus* (GM Index 1.6) was lower than *A. fumigatus* (3.1; *P* = 0.031) and the sensitivity of GM detection was lower in *A. flavus* infections [56]. A study conducted by Badiee et al. in children with IA demonstrated sensitivity and specificity of GM, (1→3)-β-d-glucan and PCR was 90% and 92%, 50% and 46%, and 80% and 96%, respectively [57]. Among ten culture positive proven and probable IA cases, *A. flavus* was identified in four samples [57]. This study showed higher levels of GM antigen in *A. fumigatus* and higher (1→3)-β-d-glucan in *A. flavus* cases [57]. However, this discrepancy in in-vitro and in-vivo GM levels reported with *A. flavus* and *A. fumigatus* warrants further studies. In patients with cerebral aspergillosis, high levels of GM antigen were detected in serum (3.4 ng/ml), which decreased to 2.8 ng/mL after therapy [58]. An animal model study by Walsh et al. showed that a pan-*Aspergillus* real-time PCR assay in BAL samples was positive in all *A. flavus* infected untreated rabbits while species-specific PCR could only detect 38% of samples. However, all other species were detected by both primers [59]. The utility and cut-off values of different biomarkers used for the diagnosis and differentiation of aspergillosis due to *A. flavus* and other *Aspergillus* spp. is the subject of further investigation. 

## 5. Outbreaks

*A. flavus* is ubiquitously distributed in air, soil, and water. However, the quantity of conidia/ spores in outdoor-air and indoor-air in the home and hospital environment are considered important for causing different forms of *Aspergillus* diseases [29]. Activities related to construction, renovation, and demolition of buildings increases the spore count and can lead to outbreaks, especially in the hospital setting [60,61]. Studies from Iran have shown that *A. flavus* is the most common *Aspergillus* isolated from hospital and home air [62,63]. Air containing conidia from construction related sources such as entry of unfiltered air, backflow of contaminated air, air filters, fireproofing materials, air conditioners, air conditioning duct systems, and dust of false ceilings can be disseminated throughout hospital areas and get aerosolized [61,64]. A study from India compared the fungal spore burden in air-conditioned and non-air-conditioned areas of the hospital and showed high spore counts in both the areas [65]. The average number of *Aspergillus* species spores were significantly higher (*p* = 0.013) than other molds in non-air-conditioned area [65]. *A. flavus* was the most common *Aspergillus* species isolated from air-conditioned areas [65]. Hospital outbreaks due to *A. flavus* have been reported mainly in immunocompromised patients and they present as invasive pulmonary, sinus, or other forms [61,66,67,68,69]. In a large review of nosocomial outbreaks of aspergillosis, *A. fumigatus* was the most common (*n* = 154 patients) species implicated followed by *A. flavus* (*n* = 101 patients) [70]. Outbreaks due to *A. flavus* have also been associated with cutaneous, mucosal, and subcutaneous tissues, whereas *A. fumigatus* is shown to cause either pulmonary or sinus disease [70]. Nosocomial outbreaks following cataract surgery have been noted mainly from India. In a large series of post-cataract endophthalmitis, Narang et al. reported 27 cases of post-cataract surgery endophthalmitis in which *A. flavus* was isolated in 59% of the cases [31]. In a review of nosocomial aspergillosis, outbreak associated environmental investigations were noted in 24/53 reports, and the spore counts varied between 0–100 spores/m^3^. Even a very low *Aspergillus* count (<1 colony forming unit/m^3^) within the hospital environment can cause infections in high-risk patients [70]. The major organs involved during fungal outbreaks in hospitals were only lung (46%) followed by lung with other sites (20%), skin/wound (7%), sinus with other sites, eye, and disseminated multi-organ disease [61]. The overall mortality rate was 58%, and the source of outbreaks in the majority of cases was attributed to construction, renovation, or demolition work in the hospital [61,70].

## 6. Taxonomy and Identification

Micheli, an Italian priest, described *Aspergillus* for the first time in 1729. He named it based on the morphological resemblance of conidial head of *Aspergillus* to the ‘aspergillum’, the holy water sprinkler [71]. Since the last decades, *Aspergillus* is classified based on molecular and chemotaxonomic characterization [72]. *Aspergillus* belongs to the Kingdom Fungi, Division Ascomycetes, class Eurotiomycetes, order Eurotiales and Family Trichomaceae. In 2014, Samson et al. provided a list of 339 species based on Internal Transcribed Spacer (ITS), calmodulin, β-tubulin, and DNA-directed RNA polymerase II subunit sequences [72]. Based on phylogenetic analysis, phenotypic, and physiological characters, *Aspergillus* is divided into six subgenera viz. *Circumdati*, *Nidulantes*, *Aspergillus*, *Fumigati, Polypaecili*, and *Cremei*. Each subgenus is divided into several sections of closely related species [2,72]. Visage et al. classified *Aspergillus* into ten clades with different sections within it [73]. 

Even before the introduction of modern tools, Raper and Fennell (1965) described nine species and two varieties in *Aspergillus* section *Flavi* using phenotypic techniques [29]. Subsequently, Hedayati et al. compiled a list of 23 species or varieties of *A. flavus* based on identification by morphological characteristics alone [29]. Using a polyphasic approach involving analysis of morphological characters, extrolite data and partial sequences of calmodulin, β-tubulin, and ITS, Varga et al. divided the *Aspergillus* section *Flavi* in 22 species from seven clades, *A. flavus* (*A. flavus*, *A. oryzae*, *A. parasiticus*, *A. minisclerotigenes*, *A. parvisclerotigenes*), *A. tamarii* (*A. tamarii, A. terricola, A. terricola var. indicus, A. flavofurticus, A. caelatus, A. pseudotamarii, A. pseudocaelatus)*, *A. nomius*, (*A. nomius, A. pseudonomius*, *A. bombycis*), *P. alliaceus* (*A. alliaceus, A. albertensis, A. lanosus*, *A. albertensis*, *A. lanosusis*) *A. togoensis* (*A. togoensis, A. coremiiformis)*, *A. leporis* (*A. leporis*, *A. coremiiformis*) and *A. avenaceus* (*A. avenaceus*, *A. coremiiformis)* [74]. Further, they identified two novel species, namely *A. pseudocaelatus* and *A. pseudonomius* [74]. Morphologically this section is characterized by its biseriate conidial heads appearing yellow-green to brown with dark sclerotia. *A. flavus* generally reproduces asexually and resides in soil either as sclerotia or conidia. The sclerotia germinate into mycelium producing numerous chains of conidia that separates, disperse, and disseminate widely into the environment [71].

*A. flavus* can grow better at an optimum temperature of 37 °C contributing to the pathogenicity, although the growth temperatures vary between 12 and 48 °C [29]. The sexual stage (teleomorph) of *A. flavus* was identified by Horn et al. and named as *Petromyces flavus* [75]. The wide diversity in *A. flavus* has been attributed to sexual reproduction and recombination [76]. *A. oryzae* is very closely related to *A. flavus*. Genome sequence data support that these two are the same species and *A. oryzae* is the domesticated variant of *A. flavus* [71]. *A. flavus* can produce harmful toxins, while *A. oryzae* is used in the food industry and for industrial enzyme production. The genome size of *A. flavus* is ~36 Mb with approximately 12,197 predicted genes [71,77]. Of the many species described in this section, *A. flavus* sensu-stricto is commonly implicated in human diseases and other rarely reported species include *A. tamarii* and *P. alliaceus* [3,14,78].

## 7. Molecular Epidemiology

Molecular typing techniques for *Aspergillus* species have been mainly evaluated in *A. fumigatus* with limited studies involving *A. flavus*. Assessing the epidemiological relationship between patient and environmental isolates by molecular strain typing may provide an insight into our understanding of the dynamics of *A. flavus* infections in humans and the environment [79]. 

### 7.1. Genotyping by Random Amplified Polymorphic DNA (RAPD)

This technique was used earlier for typing of *A. flavus* isolates. In an outbreak of nosocomial sternal surgical site infection following cardiac surgery, 15 isolates collected from the air and surfaces of the surgical ward and three clinical isolates (two from surgical site infection and one from bronchial aspiration) were typed by RAPD and it was found that all three clinical isolates and one environmental isolate had the same RAPD type suggesting hospital-acquired infection from a single contaminated source [80]. It was also demonstrated that *A. flavus,* isolated from a case of nosocomial infection that underwent cardiac surgery, shared the same RAPD type with environmental isolates collected from the grilles of a dual reservoir cooler-heater [81]. In a small outbreak of *A. fumigatus* and *A. flavus* in the hematology unit of Erasmus MC in Rotterdam, The Netherlands, genotyping of *A. flavus* and other *Aspergillus* isolates was performed by RAPD and it was proven that an outbreak of IA in the hematology ward was due to unrelated events in the hospital and was not due to a common source within the hospital [82]. Genotyping by RAPD may help to identify outbreaks of nosocomial origin in spite of limited reproducibility of patterns (variation of number, size, and intensity of bands) and inter-laboratory reproducibility [76,83].

### 7.2. Restriction Fragment Length Polymorphisms (RFLP)

Molecular typing using RFLP of mitochondrial DNA can not only efficiently differentiate different species within the *A. flavus* species complex, but it can also differentiate between two very closely related species, *A. flavus* and *A. oryzae* [84]. PCR-RFLP using 11 nuclear genes and the restriction enzyme, *HaeIII* followed by DNA sequencing has been found a useful method for screening nucleotide polymorphisms in *A. flavus* isolates [85]. RFLP applied to investigate a presumed outbreak of *A. flavus* in seven immunocompromised pediatric patients with IA showed that there was no single source of the infection except the sharing of the same fingerprint among isolates from a health care worker, one patient and an environmental strain [86]. 

### 7.3. Single Strand Confirmation Polymorphism (SSCP)

Other molecular typing techniques such as DNA fingerprinting using the pAF-28 probe and PCR-single strand confirmation polymorphism (SSCP) using ITS 1 and 4 has been demonstrated to possess high reproducibility and discriminatory power for tracking the origin of *A. flavus* infections and also to differentiate species within the *A. flavus* species complex [87,88]. 

### 7.4. Amplified Fragment Length Polymorphism (AFLP)

AFLP is considered a highly discriminatory and reproducible technique that helps in differentiating fungi to the strain level. This technique is advantageous as fragments covering the whole genome are screened, and prior information of the genome sequence is not essential. Montiel et al. used AFLP using 12 different primer combinations to differentiate 24 isolates belonging to *Aspergillus* section *Flavi* complex (*A. flavus*-8, *A. oryzae*-7, *A. sojae*-6, and *A. parasiticus*-3) [89]. Their technique clearly separated *A. sojae/A. parasiticus* from *A. oryzae/A. flavus*. Further, the primers could differentiate *A. sojae* from *A. parasiticus* but not *A. oryzae* from *A. flavus* [89]. Rudramurthy et al. evaluated a large collection of clinical *A. flavus* isolates by AFLP and showed its high discriminatory power, equivalent to multi-locus microsatellite typing in differentiating *A. flavus* isolates into different clades [90]. 

### 7.5. Multilocus Microsatellite Typing (MLMT)

MLMT is a rapid and relatively inexpensive typing method capable of amplifying short repetitive sequences abundantly present in the genome. This technique was first applied to type *A. flavus* isolated from cotton [91]. Using 24 microsatellites, a high degree of genetic variability in *A. flavus* strains was demonstrated [91]. Later, Rudramurthy et al. used nine multiplex multicolor microsatellite panels for genotyping 149 clinical *A. flavus* isolates collected from cases with different clinical conditions and found 124 genotypes with excellent discriminatory power and high reproducibility [90]. The overall discriminatory power of the technique was 0.997, of which two markers, 2A (0.954) and 3B (0.944), were recognized as the highest discriminatory markers. There was no correlation between genotype and clinical disease. [90]. Recently, the same nine markers [90] were used to evaluate clinical (*n* = 121) and environmental (*n* = 79) isolates originating from different parts of Iran [92]. It was concluded that clinical isolates were unrelated as all the strains had unique genotypes. However, they found clustering of one clinical strain with two environmental strains isolated from the same hospital suggesting hospital-acquired infection or colonization [92]. Similarly, using the same nine microsatellite panels, 49 *A. flavus* isolates were separated into 36 different genotypes. [93]. In another study, by using a combination of six markers, a total of 48 different genotypes of *A. flavus* were identified from patients with invasive aspergillosis with good genetic diversity within isolates colonizing the patients [94]. Guarro et al. reported a nosocomial outbreak of IA due to *A. flavus* in a general medical ward by performing microsatellite-based molecular typing of 28 isolates collected from the hospital environment and patients [95]. They showed a genotype shared by three clinical isolates, while two clinical isolates clustered separately. In both groups of isolates, all three isolates were spatially and temporally related [95]. Wang et al. also evaluated a multi-locus variable number tandem repeat analysis (MLVA) technique in *A. flavus* isolates which showed similar findings [96]. These findings suggest that any *A. flavus* strain may be potent enough to cause any clinical variety of aspergillosis depending on host factors and exposure. Disease-specific genotypes which cause a particular type of aspergillosis possibly do not exist in *A. flavus.*


### 7.6. Multi-Locus Sequence Typing (MLST)

Although multi-locus sequence typing (MLST) has primarily gained importance to understand the epidemiology of bacterial pathogens until now. MLST scheme has not been described or applied for typing *A. flavus*. Few multi-locus gene sequences have been used to delineate different species within *Aspergillus* section *Flavi*, but it has not been applied to type *A. flavus sensu-stricto* or its other closely related species. However, for *A. flavus*, microsatellite typing performs better in terms of discriminatory power and thus can be preferred over other typing techniques [97]. 

## 8. Antifungal Susceptibility and Wild Type Distribution

Though the magnitude and mechanism of azole resistance in *A. fumigatus* is well studied, [98,99,100,101] there are only a few reports available for *A. flavus*. It is considered that the true incidence of resistance in *Aspergillus* species might be much higher than currently known and thus, routine in-vitro susceptibility testing of all clinical *Aspergillus* isolates is recommended [102]. In a large screening study of 1789 *Aspergillus* isolates, Pfaller et al. showed that the frequency of non-wild type (non-WT) *A. flavus* isolates for itraconazole, voriconazole and posaconazole were 0.8%, 1.7%, and 5.1% respectively [103]. In a collection of clinical and environmental (*n* = 188) *A. flavus* isolates from a center in India, a total of 5% of the isolates were found to be non-WT to azoles [104]. Another report from India found 2.5% azole resistance in *A. flavus* [105]. In a surveillance study of a transplant unit, none of the *A. flavus* isolates was resistant to any triazole tested [106]. In comparison to *A. fumigatus*, resistance to itraconazole and voriconazole in *A. flavus* is very rare although both species are exposed to the same azole fungicides in the environment. Wathiqi et al. demonstrated that the mean MIC values for voriconazole and posaconazole of environmental strains were lower than clinical isolates [78]. Similarly, Araujo et al. showed lower MIC values of environmental strains for itraconazole than clinical isolates [107]. The Epidemiological Cut-off Values (ECVs/ECOFFs) and wild type distributions of MIC/MEC’s have been defined by various studies using CLSI/EUCAST broth microdilution, Sensititre Yeast One (SYO), and E-test (Table 1). Comparison of EUCAST wild type MIC distributions for different antifungals between *A. fumigatus* and *A. flavus* is provided in Figure 1. Espinel-Ingroff et al. defined ECV-WT comprising ≥95% of the modeled MIC population against voriconazole for *A. flavus* as ≤1 µg/mL [108]. In a collection of 590 clinical isolates, from five centers in USA and Europe, the rate of voriconazole resistance in *A. flavus* was estimated at ~2% using an ECV of >1 µg/mL [109]. Lalitha et al. demonstrated that 12.5% of ocular *A. flavus* isolates were non-WT for voriconazole [110]. MICs of isavuconazole for *A. flavus* in various studies showed better activity [111] or lower activity [112] than voriconazole. 

## 9. Antifungal Resistance

### 9.1. Mutations in the Cyp51 Gene and Azole Resistance

Limited attempts have been made to evaluate the azole resistance mechanism in *A. flavus*. Similar to *A. fumigatus*, studies have been done to investigate mutations in the gene encoding lanosterol 14-alpha-demethylase. A voriconazole resistant clinical *A. flavus* isolate reported from China harbored a mutation at T788G (S240A) in the *cyp51C* allele [120]. Krishnan-Natesan et al. reported mutations in *cyp51A* and *cyp51B* in 40% of in-vitro selected voriconazole resistant isogenic *A. flavus* isolates [121]. Recently, a clinical voriconazole resistant isolate with Y319H mutation in the *cyp51C* allele of lanosterol demethylase coding gene was demonstrated [122]. Sharma et al. demonstrated polymorphisms in azole target genes (*cyp51A*, *cyp51B*, and *cyp51C*) in three non-WT *A. flavus* isolates exhibiting high voriconazole MICs using whole-genome sequence analysis. They also demonstrated four novel substitutions (S196F, A324P, N423D, and V465M) in the *cyp51C* gene in one of those non-WT isolates [105]. Both of the above studies demonstrated that mutations impart structural and functional changes interfering with binding affinity of the enzyme [105,122]. Choi et al. sequenced the target genes (*cyp51A, cyp51B*, and *cyp51C*) and their promoter regions in 50 clinical isolates of *A. flavus* (including seven voriconazole non-WT isolates) originated from South Korea [93]. They confirmed the findings of the other studies [105,122] that S240A in *cyp51C* was present in all *A. flavus* isolates irrespective of their voriconazole susceptibility. The non-synonymous mutations in the *cyp51C* gene (S196F, A324P, N423D, and V465M) described by Sharma et al. were present in all seven of the non-WT *A. flavus* and 28.6% (12/42) of the wild type isolates [93].

### 9.2. Azole Resistance and Gene Expression

Differential expression of ATP binding cassettes (ABC) and major facilitator superfamily (MFS) transporters are the other reported mechanisms of acquired resistance in *A. flavus* [123,124,125]. However, some of these mechanisms were found under in-vitro selection conditions, which necessarily may not mimic the in-vivo selection of azole resistance. In a recent study from India, basal level and voriconazole-induced expression of *cyp51A* homologs and various efflux pump genes were analyzed in three each of non-WT and wild-type *A. flavus* isolates [104]. Irrespective of the azole susceptibility of the isolates, a low basal expression of all the efflux pumps was noted. However, the non-wild-type isolates demonstrated heterogeneous upregulation of efflux pumps and target enzyme-coding genes after induction with voriconazole. A dichotomy in the induction of *Cdr*1B expression was observed between the resistant and susceptible isolates. In two resistant isolates, an eight–nine-fold increase in the *Cdr*1B gene following induction compared to the susceptible reference strain [104]. Interestingly, overexpression of *cdr1B*, a homolog of cdr1 efflux pump in *Candida albicans* has been reported earlier in azole resistance isolates of *A. fumigatus* which were lacking target site mutation [126]. Two non-WT isolates from the same patient also demonstrated greater overexpression of *Mdr1*, *Mdr2*, *Mdr4*, and *atrF* [104]. Similarly, the overexpression of *Mdr1*, *Mdr2*, *atrF*, and *mfs1* genes were reported by Sharma et al. and *cyp51A*, *Mdr2*, *Mfs1*, and *atrF* were reported by Choi et al. in non-WT *A. flavus* [93,105]. These results suggest that overexpression of efflux pumps and target genes in those non-WT strains lacking mutations in the *cyp51* gene can lead to high voriconazole MICs in clinical *A. flavus* isolates. It has also been reported that a mutation in *yap1* (Leu558Trp) gene, the transcription factor involved in oxidative stress, is responsible for a voriconazole resistance phenotype which acts by upregulation of *atrF* transporter [127]. 

### 9.3. Azole Resistance and Exposure to Environmental Fungicides

In *A. fumigatus*, the most common way of acquiring resistance to azoles has been traced to the environment due to unintended exposure of *A. fumigatus* to azole fungicides [98,99]. Unlike *A. fumigatus*, a dominant marker of resistance is still elusive in *A. flavus*. The pursuit for such a marker has largely been precluded by the limited number of azole-resistant *A. flavus* isolates available for analysis. Although *A. flavus* could similarly get exposed to fungicides in the environment and may evolve resistance, such a phenomenon has not been reported yet. Sexual reproduction has been reported to occur under laboratory conditions, but it remains to be seen how frequently *A. flavus* reproduces sexually. A comparative study of different *Aspergillus* species on sexual reproduction, genomic plasticity, and evolution of resistance may provide hints regarding the lower tendency of azole resistance in *A. flavus.* However, detailed environmental surveillance studies of azole resistance in *A. flavus* have not been carried out, and this requires investigation.

### 9.4. Polyene Resistance

*A. flavus* exhibits variable MICs to amphotericin B. A study by Rudramurthy et al. showed high MICs of ≥2 μg/mL in the majority (91.8%) of *A. flavus* isolates with a geometric mean MIC of 3.52 μg/mL and MIC_90_ of 8 μg/mL [116]. Other studies found 66.6% [128], 84% [129,130], and 87% [131] reduced susceptibility to amphotericin B. Lass-Floerl et al. also showed that 67% of *A. flavus* isolates in Austria with high MIC to amphotericin B were associated with amphotericin B therapy failure [128]. These reports support that *A. flavus* may be intrinsically resistant to amphotericin B. Detection of an amphotericin B resistant strain of *A. flavus* isolated from a cave in Brazil having a MIC value of 4 μg/mL supports this notion [132]. Though mechanisms of amphotericin B resistance is not known in *A. flavus*, it appears that it may be due to higher ergosterol levels and increased enzymatic activity of the peroxidase and superoxide dismutase, with lower lipid peroxidation [132]. 

## 10. Clinical Management of Aspergillosis Due to *A. flavus*

Management of invasive aspergillosis includes use of extended-spectrum triazoles [102]. Treatment of *A. flavus* infections is similar to that caused by other species of *Aspergillus* [29]. In the absence of defined clinical breakpoints for all the different antifungal agents against *A. flavus*, many studies have resorted to epidemiological cut-off values (ECV/ECOFF) to assist in the management of *A. flavus* infection. EUCAST has defined the breakpoints for itraconazole (2 μg/mL), isavuconazole (1 μg/mL), amphotericin B (2 μg/mL), and posaconazole (0.25 μg/mL) against *A. fumigatus* while in *A. flavus*, breakpoints for itraconazole only are available. However, the committee has proposed ECOFF values for isavuconazole, itraconazole, and voriconazole for both *A. fumigatus* and *A. flavus* [133,134] (Table 2). ECV/ECOFF of EUCAST is generally the same or one-fold higher than that defined by CLSI [135] (Table 2).

Among triazoles, voriconazole is the preferred drug of choice to treat all forms of invasive aspergillosis. For those who receive a prolonged course of triazoles, therapeutic drug monitoring is recommended which helps to improve the therapeutic efficacy, assess therapeutic failures due to suboptimal exposure and reduce toxicity associated with azoles [136,137]. No clinical trials are available evaluating the clinical efficacy of voriconazole specifically for aspergillosis due to *A. flavus*. A study evaluating the efficacy of voriconazole in a non-neutropenic murine model of disseminated *A. flavus* infection using two voriconazole non-WT isolates (one harboring the Y319H substitution in the *cyp51C* gene) and two wild-type isolates showed a dose-response relationship with improved mouse survival in a dose-dependent manner with all isolates [138]. Increasing doses increased the survival of the mice in a dose-dependent manner. However, overall, the AUC and AUC/MIC ratio showed a better exposure–survival relationship. The interdependence between MIC, mutation, and overall effect showed that lower exposures were required for strains with higher MICs to result in the same effect indicating the underlying mutations in the target gene may have a significant impact on the dose or exposure required for the treatment of infection [138].

In *A. flavus,* posaconazole has been found to inhibit the azole target enzyme more efficiently than voriconazole and itraconazole [3]. Shivaprakash et al. also reported better in-vitro anti-*A. flavus* activity of posaconazole than itraconazole, isavuconazole, and voriconazole [111]. Further, they observed all (*n* = 188) *A. flavus* isolates exhibiting MICs of ≤1 µg/mL for itraconazole and posaconazole while 99.5% and 74% isolates had MICs of ≤1 µg/mL for isavuconazole and voriconazole, respectively [111]. Posaconazole has been licensed for use as prophylaxis in certain immunosuppressed patients in the USA and Europe for invasive aspergillosis refractory to amphotericin B formulations or itraconazole [136]. Isavuconazole has been approved by the Food and Drug Administration for the treatment of invasive aspergillosis and mucormycosis [139]. Isavuconazole was found to be non-inferior to voriconazole in the treatment of invasive pulmonary aspergillosis (IPA), but it is better tolerated with fewer drug-related adverse events [140]. 

Patients with invasive sino-nasal aspergillosis due to *A. flavus*, as seen in Sudan and other tropical countries, tend to have a more indolent progression over months to years. For these patients, the Infectious Diseases Society of America (IDSA) recommends treating them aggressively with combined surgical debridement and long-term antifungal therapy for at least one year to prevent recurrent infection [136,137]. Further, MIC values and evaluation of pharmacodynamics (in an in-vivo and in-vitro model of invasive fungal sinusitis) of F901318 (olorofim) have shown this drug as a potential new agent for the treatment of invasive infections caused by *A. flavus* and azole-resistant *A. fumigatus* [141,142,143]. Fungal balls of the lung may rarely be caused by *A. flavus,* and the course of the infection is generally not rapidly progressive hence acute management is essential only if the lesion worsens as noticed with the occurrence of hemoptysis [137].

Echinocandins are an important group of fungistatic drugs [3,144]. An in-vivo study of *A. flavus* infected mice has shown combination therapy of anidulafungin and voriconazole to be more effective than anidulafungin alone or in some cases better than voriconazole alone. It reduced the fungal load in tissues as well as galactomannan level in the serum of infected mice [144]. IDSA recommends a combination of an echinocandin with voriconazole (weak) for a select group of patients with documented IPA and does not recommend (strong) primary therapy with echinocandins alone [137]. However, echinocandins may be used in situations when azoles and amphotericin B are contraindicated [137]. As echinocandins poorly penetrate the blood-brain barrier, they cannot be used for cerebral aspergillosis. [102,137].

Amphotericin B deoxycholate and its lipid derivatives are appropriate options for initial and salvage therapy of *Aspergillus* infections when voriconazole cannot be administered [102,137]. Because *A. flavus* is generally shown to have reduced susceptibility to amphotericin B compared to *A. fumigatus* [116,128,129,130,131], the European Society for Clinical Microbiology and Infectious Diseases strongly recommends avoiding amphotericin B for aspergillosis caused by *A. flavus* species complex [102]. 

Though ECVs have demonstrated the emergence of drug-resistant strains, triazoles are still preferred in the management of the majority of IA cases due to *A. flavus*. ECV is thus, an important part of the routine resistance surveillance program to detect the emergence of strains with decreased susceptibility to a particular antifungal agent. 

## 11. Conclusions

Worldwide, *A. flavus* is an important causative agent of invasive aspergillosis with a higher incidence in tropical countries. Invasive rhinosinusitis and pulmonary forms are the commonest presentations of invasive aspergillosis due to *A. flavus*. The utility and significance of cut-off values of different biomarkers used for the diagnosis and differentiation of aspergillosis due to *A. flavus* and other *Aspergillus* species is the subject of further investigation. This agent can cause hospital outbreaks, especially after surgery in high-risk patients. Multilocus microsatellite typing is the most discriminatory typing technique that may help in source determination during hospital outbreaks. Triazole resistance, though rare, has been reported with this pathogen. *A. flavus* exhibit high amphotericin MICs, and there is a need to unravel the mechanism of resistance to this antifungal. A fungicide driven route of acquiring azole resistance with this agent is possible but needs further studies. Resistance breakpoints are available only for itraconazole and for other azoles and amphotericin the ECV/ECOFF value is one step higher than *A. fumigatus*. Voriconazole is the drug of choice for treatment, and amphotericin B should be avoided. Echinocandins may be used in combination with voriconazole in selected patients or alone in a situation where azoles and amphotericin B are contraindicated. Clinical trials and pharmacodynamic studies are essential to determine breakpoints and optimize the dosage but, in the absence, ECV/ECOFF values can be used as a guide in choosing an appropriate antifungal agent for therapy. 

## Figures and Tables

**Figure 1 jof-05-00055-f001:**
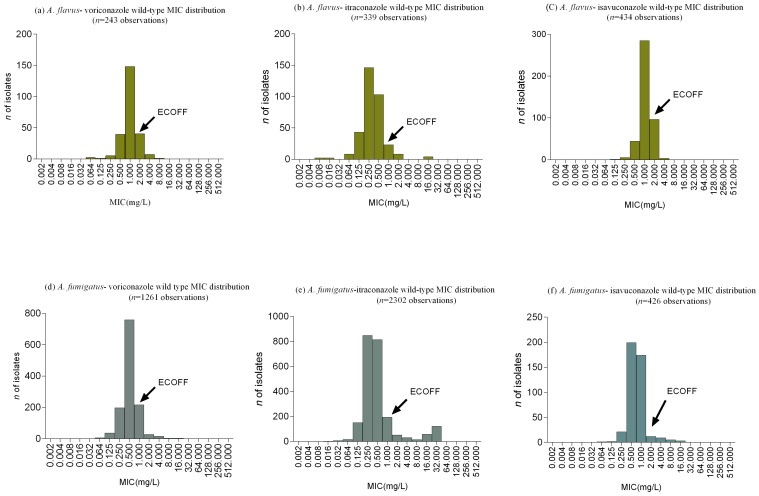
Comparison of European Committee on Antimicrobial Susceptibility testing wild-type azole minimum inhibitory concentrations distribution of *A. flavus* and *A. fumigatus.*

**Table 1 jof-05-00055-t001:** Review of Epidemiological Cut-off Values (ECVs/ECOFFs) of various antifungal agents against *Aspergillus flavus* isolates.

Study (Reference)	Testing Method	Number of Isolates	Antifungal	MIC Range	Modal MIC	ECV/ECOFF (95%/97.5%/99%)	% Above ECV/ECOFF (95%/97.5%/99%)	Comments
				Values in μg/mL				
Espinel-Ingroff et al. [112]	CLSI -BMD	444	ISA	0.06–2	0.5	1.0/1.0/2	3.2/3.2/0.2	Modal MIC more than POS and VOR
Espinel-Ingroff et al. [108]	CLSI -BMD	793	AMB		1	≤2		
Espinel-Ingroff et al. [109]	CLSI -BMD	536	ITR	0.03–2	0.5	1	0.4	ITR EUCAST R-BP >2
		321	POS	≤0.03–2	0.06	0.25	5.3	
		590	VOR	0.06–4	0.5	1	0.9	
Espinel-Ingroff et al. [113]	CLSI BMD	432	CAS			0.5	1.6	
Espinel-Ingroff et al. [114]	Combined CLSI EUCAST	793	AMB	0.032–8	1	2/4/4		
Pfaller et al. [103]	CLSI BMD	538	ITR	0.03–≥4	0.5	1/-/-	0.4/-/-	
		444	POS	≤0.03–≥4	0.06	0.5/-/-	0.9/-/-	
		592	VOR	0.06–≥4	0.5	1/-/-	1.7/-/-	
Jiwa et al. [115]	CLSI -BMD	50	AMB	0.12–2	0.5	2	0	All isolates are from multiple centers from Canada
			POS	0.06–1	0.25	0.25	28	
			VOR	0.5–16	1	1	40	
			ITR	0.12–2	0.5	1	2	
			CAS	0.03–0.5	0.125	0.25	4	
Rudramurthy et al. [116]	CLSI and EUCAST	208	AMB	1.0–16		-/-/16	-/-/0	All isolates are from India
			ITR	0.12–1		-/-/0.5	-/-/1.4	
			VOR	0.25–4		-/-/4	-/-/0	
			POS	0.12–0.5		-/-/0.5	-/-/0	
			ISA	0.2–4		-/-/2	-/-/1.1	
			CAS	0.25–1		-/-/1	-/-/0	
			ANI	0.008–0.016		-/-/0.016	-/-/0	
			MFG	0.008–0.2		-/-/0.125	-/-/1.4	
Paul et al. [104]	CLSI	189	ITR	0.03–16		1	1.6	Included both clinical (*n* = 121) and environmental (*n* = 68) isolates
			VOR	0.03–8.0		1	3.17	
			POS	0.015–0.5		0.25	0.5	
Espinel-Ingroff et al. [117]	SYO	389	VOR	0.008–≥16	0.25	1		
	E-test	250	ITR	0.01–2	0.25	1		
	E-test	257	VOR	0.01–≥16	0.25	0.5		
	E-test	204	POS	0.01–1	0.25	0.5		
Pfaller et al. [118]	CLSI BMD	188	CAS	0.007–0.12	0.016	0.06	0.5	
Espinel-Ingroff et al. [119]	E-Test CLSI EUCAST	238	AMB	0.12–32	2	8 (E-test) 4 (CLSI and EUCAST)		
Taghizadeh-Armaki et al. [92]	EUCAST	200	AMB	1–16			4.0	
			ITR	0.031–4			1.5	
			VOR	0.063–2			0	
			POS	0.031–1			0.5	
			ISA	0.125–4			2	

Abbreviations: ECV/ECOFF- epidemiological cut-off value; ISA-isavuconazole; POS-posaconazole; VOR -voriconazole; AMB-amphotericin B; ITR-itraconazole; CAS-caspofungin; ANI-anidulafungin; MFG-micafungin; BP-breakpoint.

**Table 2 jof-05-00055-t002:** Epidemiological cutoff values (μg/mL) defined for *Aspergillus flavus* and *A. fumigatus* by Clinical Laboratory Standard Institute (CLSI) [135] and European Committee on Antimicrobial Susceptibility Testing (EUCAST) [133,134].

Antifungal	EUCAST		CLSI	
	*A. flavus*	*A. fumigatus*	*A. flavus*	*A. fumigatus*
**Itraconazole**	1.0	1.0	1.0	1.0
**Voriconazole**	2.0	1.0	2.0	1.0
**Posaconazole**	0.5	0.25	0.5	-
**Isavuconazole**	2.0	2.0	1.0	1.0
**Caspofungin**	-	-	0.5	0.5
**Amphotericin B**	4.0	1.0	4.0	2.0

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
