# Peer review of "Invasive Aspergillosis by Aspergillus flavus: Epidemiology, Diagnosis, Antifungal Resistance, and Management"

_jof, 2019, doi:10.3390/jof5030055_

Round 1
Reviewer 1 Report
The manuscript jof-529294 is outstanding. It is the only manuscript from the several I reviewed recently that can be published in its current form without revisions.
Author Response
Comment: The manuscript jof-529294 is outstanding. It is the only manuscript from the several I reviewed recently that can be published in its current form without revisions.
Response: We are thankful to the reviewer for these kind words.
Reviewer 2 Report
This is a comprehensive and well written review on Invasive aspergillosis by Aspergillus flavus.
There are only a few remarks to make:
In Diagnosis, lines 208-226, it appears that mainly two different parameters are discussed, GM production and sensitivity of GM. As these parameters are distinct, they should be orderly separated from each other in the text, in order to facilitate the reading.
Further, the manuscript needs a careful editing as there are several (minor) language errors and typos such as subject-verb disagreements (singular/plural) etc.
Some suggestions for improvement:
In Taxonomy and identification, lines 286-287, the sentence needs to be reformulated.
Line 292: " implicated in human disease."
Line 431: "wild-type A. flavus isolates."
Lines 508-509: "Patients with invasive sino-nasal aspergillosis …, tend to have a more indolent progression…"
Line 370: "…within Aspergillus section Flavi complex, but it has not been…"
Lines 512-513:" …MIC values and evaluation of pharmacodynamics (in in-vivo and in-vitro model of invasive fungal sinusitis) of F901318 (olorofim) have shown…"
Lines 533-536: This sentence needs also to be reformulated.
Author Response
Comment: This is a comprehensive and well-written review on Invasive aspergillosis by Aspergillus flavus.
Response: We thank the reviewer for the comments and helpful suggestions.
Comment: In diagnosis, lines 208-226, it appears that mainly two different parameters are discussed, GM production and sensitivity of GM. As these parameters are distinct, they should be orderly separated from each other in the text, in order to facilitate the reading.
Response: As suggested we have now realigned the points and changed the reference accordingly.
Comment: Further, the manuscript needs careful editing as there are several (minor) language errors and typos such as subject-verb disagreements (singular/plural), etc.
Response: Edited for the entire manuscript considering all the points suggested.
Comments: In Taxonomy and identification, lines 286-287, the sentence needs to be reformulated.
Response: The sentence has been modified to make it more clear.
Comment: Line 292: " implicated in human disease."
Response: Modified in the revised version as suggested.
Comment: Line 431: "wild-type A. flavus isolates."
Response: Modified in the revised version as suggested
Comment: Lines 508-509: "Patients with invasive sino-nasal aspergillosis …, tend to have a more indolent progression…"
Response: Modified in the revised version as suggested
Comment: Line 370: "…within Aspergillus section Flavi complex, but it has not been…"
Response: Modified in the revised version as suggested
Comment:Lines 512-513:" …MIC values and evaluation of pharmacodynamics (in an in-vivo and in-vitro model of invasive fungal sinusitis) of F901318 (olorofim) have shown…"
Response: Modified in the revised version as suggested
Comment: Lines 533-536: This sentence needs also to be reformulated.
Response: The sentence has been rephrased.
Reviewer 3 Report
In this review, the authors gave an overview of the epidemiology, clinical spectrum, diagnosis and antifungal drug resistance related to the species Aspergillus flavus. it is among the first review on A.flavus including all these aspects and it has been exhaustively and well documented.
Minor points to be consider: The authors specify that "A.flavus is the cause of a broad spectrum of human diseases predominantly in Asia, the middle east and Africa due to its ability to survive better in hot and arid climatic conditions". It will be interesting to develop more on the biology of A. flavus for example in comparison to A.fumigatus in order to explain such differences in terms of geographical distribution.
Author Response
Comments: The authors specify that "A. flavus is the cause of a broad spectrum of human diseases predominantly in Asia, the middle east and Africa due to its ability to survive better in hot and arid climatic conditions". It will be interesting to develop more on the biology of A. flavus for example in comparison to A. fumigatus in order to explain such differences in terms of geographical distribution.
Response: We highly appreciate the reviewer for raising this important query. The most available information attributes climatic variations as a reason for the difference in the geographical distribution of Aspergillus species. To the best of our knowledge, a comparison of A. flavus and A. fumigatus biology in relation to geographic distribution has not been conducted. A sentence pointing this out has been included in the revised manuscript.